# The Effect of Radial Extracorporeal Shock Wave Therapy (rESWT) on the Skin Surface Temperature of the Longissimus Dorsi Muscle in Clinically Healthy Racing Thoroughbreds: A Preliminary Study

**DOI:** 10.3390/ani13122028

**Published:** 2023-06-18

**Authors:** Karolina Śniegucka, Maria Soroko-Dubrovina, Paulina Zielińska, Krzysztof Dudek, Kristína Žuffová

**Affiliations:** 1Institute of Animal Breeding, Wroclaw University of Environmental and Life Sciences, 51-630 Wroclaw, Poland; karolina.sniegucka@upwr.edu.pl; 2Department of Surgery, Wroclaw University of Environmental and Life Sciences, 50-366 Wroclaw, Poland; paulina.zielinska@upwr.edu.pl; 3Center for Statistical Analysis, Wroclaw Medical University, 50-368 Wroclaw, Poland; 4Equine Clinic, Brno University of Veterinary and Pharmaceutical Sciences, 612-42 Brno, Czech Republic; zuffovak@vfu.cz

**Keywords:** radial extracorporeal shock wave therapy, longissimus dorsi muscle, physiotherapy, thermography, skin surface temperature, racehorses, thoroughbreds

## Abstract

**Simple Summary:**

Radial extracorporeal shock wave therapy (rESWT) is increasingly being used to treat musculoskeletal injuries in horses. The objective of this preliminary study was to assess the influence of rESWT on the skin surface temperature of the longissimus dorsi muscle of clinically healthy thoroughbreds. The horses were divided into a study group and an rESWT-sham group. A single round of rESWT was performed on the longissimus dorsi muscle. The rESWT-sham group underwent therapy without activation of the probe. In both groups, thermographic examination was performed before and just after rESWT to assess changes in skin surface temperature. In addition, thermographic examination was performed 10 min after rESWT. Palpation examination was performed after the first and second thermography examination to asses changes in muscle tone. In both groups, there was an increase in skin surface temperature just after rESWT and a decrease 10 min after rESWT. In the study group, the skin surface temperature just after rESWT was higher than in the rESWT-sham group. In the study group, the average muscle tone before the rESWT was significantly higher than just after the procedure, whereas in the rESWT-sham group the change was not significant. This study proved that rESWT increases skin surface temperature of the longissimus dorsi muscle in clinically healthy horses. Further research is necessary in order to design shockwave treatment with appropriate parameters for effective and safe therapy.

**Abstract:**

Radial extracorporeal shock wave therapy (rESWT) is increasingly being used to treat musculoskeletal injuries in horses. The aim of this study was to assess the influence of rESWT on the skin surface temperature of the longissimus dorsi muscle in clinically healthy racing horses. A total of 24 thoroughbreds were divided into a study group (*n* = 12) and an rESWT-sham group (*n* = 12). The study group underwent rESWT, whereas the rESWT-sham group had rESWT without probe activation in the treated area. Both groups underwent thermographic examination before and just after rESWT to determine and compare skin surface temperatures. Palpation examination was performed after the first and second thermography examination to assess longissimus dorsi muscle tone. Additionally, thermographic examination was repeated 10 min after the rESWT. In both groups, there was an increase in skin surface temperature just after rESWT, and a decrease 10 min after it to below the initial value. In the study group, the skin surface temperature just after rESWT was higher than in the rESWT-sham group. Additionally, in the study group the average muscle tone before rESWT was significantly higher than just after the procedure, whereas in the rESWT-sham group the average change in muscle tone was not significant. The results proved that rESWT increases skin surface temperature of the longissimus dorsi muscle in clinically healthy horses. Further research is necessary in order to configure shockwave treatment with appropriate parameters for effective and safe therapy.

## 1. Introduction

Extracorporeal shock wave therapy (ESWT) is a treatment method for musculoskeletal injuries which has been gaining popularity in equine veterinary medicine [1]. The method utilizes a mechanical wave generated outside the body [2]. The waves are described in terms of specific physical parameters, including a rapid rise time, high peak pressures and a gradual decrease in pressure over a few milliseconds [3]. Treatment consists of pressure in the range of 10 to 100 MPa, a fast rise time from 30 to 120 ns and a short pulse duration: 5 μs [4]. After the rapid increase in pressure, there is a longer period of pressure decrease until it returns to normal and then becomes negative [2,4]. The level of energy, the pulse frequency, the type of tissue through which the wave propagates and the penetration depth all depend on the wave parameters [2]. The shock wave pressure generated translocates through fluid and soft tissue, with its effects appearing in places where there is a change in tissue density, such as the boundary between bone and soft tissue [5]. Shock waves can propagate through tissues of similar acoustic impedance such as fat, muscle and water [2].

There are two types of ESWT, which differ according to the way the wave spreads: radial shock wave therapy (rESWT) and focused shock wave therapy (fESWT) [6]. In the first type, the radiation wave propagates through the tissues radially, with the pressure building up during the course of wave generation to a maximum of 10 Mpa at the source. In the second type, the focused wave generates a pressure field which converges at an adjustable focus, at determined depths within tissue where maximum pressure is achieved [6]. While rESWT gives a more superficial effect, fESWT can penetrate to twice the depth of rESWT [7]. The depth of penetration of radial waves is 3 to 3.5 cm in human tissue [8,9,10,11]. In another study, a penetration depth of well over 10 cm could be achieved with focused waves [12]. The radial wave has slower rise times and lower peak pressures than the focused wave, which may be a cause of differences in the effects on the treated tissue [3]. However, significant clinical differences between rESWT and fESWT have not been confirmed [2].

The influence of rESWT on biological tissue response has been documented in a number of studies. Zhang et al. [13] have indicated that the properties of rESWT significantly promoted the proliferation and self-renewal of human bone marrow mesenchymal cells in vitro and accelerated the cartilage repair process in vivo, using a rabbit model. Another study, on fetal rat metatarsal bones, proved that rESWT can stimulate growth plate chondrogenesis and longitudinal bone growth [14]. Mattyasovszky et al. [15] have demonstrated that rESWT has the potential to modulate the biological function of human skeletal muscle cells influenced by gene expression and myogenic factors. However, an in vitro study based on equine cartilage explants has shown no significant effect on the degeneration of glycosaminoglycans, or the synthesis of nitric oxides and prostaglandin (E2), following rESWT [16].

The main advantages of rESWT are that the absorption of the shock wave contributes to analgesia, pain reduction and a decrement in the muscle tone of the treated tissue. A study presented by McClure et al. [2] detected cutaneous analgesia of the overlying skin up to the 3rd metacarpal bone during the first 72 h after application of ESWT in horses. In a study based on patients affected by multiple sclerosis, a series of rESWT sessions contributed to pain and muscle tone decreases [17]. Vidal et al. [18], who studied fifteen patients with spastic cerebral palsy, found a significant increase in range of motion after rESWT, proving the effect of muscle tone reduction. The positive results were observed for at least two months after treatment. In another study, a single treatment session of rESWT decreased the muscle tone of the spastic plantar flexor muscles of the feet in children with cerebral palsy [19]. All patients received a single session of rESWT in their gastrocnemius and soleus muscles. Additionally, gastrocnemius muscle tone in hemiplegic stroke patients was found to significantly improve the spasticity of the ankle plantar flexor after rESWT [20]. One session of rESWT was applied on the medial head of the gastrocnemius, although there were no significant effects of rESWT on the amplitude of tibial nerve conduction.

Previous studies have indicated a neovascularization effect of rESWT [21]. A study on the rat model of cerebral ischemia demonstrated that rESWT increased cerebral blood flow and the expression of vascular endothelial growth factor, while reducing brain infarct volume [22]. Contaldo et al. [23] showed improved wound healing in murines after rESWT, which was associated with an increasing functional neovascular density. Microcirculation of the wound was determined quantitatively, in vivo, with epi-illumination intravital fluorescence microscopy. An increase in functional angiogenetic density was observed on day 5 (23%), day 7 (36%) and day 9 (41%). However, in a study on healing the wounds of the distal part of the forelimb in horses, neovascularization after rESWT was not detected [24]. The authors have suggested that the intensity of rESWT may have been inadequate for inducing increased neovascularization. Another study demonstrated that rESWT can improve skin flap survival rate, through enhanced vasodilatation and neovascularization, via modulation of angio-active factor expression [25].

In equine veterinary medicine, rESWT has been used to treat chronic insertion desmopathy of the proximal suspensory ligament [26,27]. Other studies have demonstrated the applicability of rESWT in the treatment of navicular syndrome [28]. It has also been reported that rESWT can be used safely in the treatment of dorsal metacarpal disease in thoroughbreds [29]. This ex vivo experiment showed significant effects of fESWT and rESWT on 11 racing thoroughbreds and 5 racing horses with third metacarpal and third metatarsal bone microdamage. The author observed that fESWT was associated with increased microcrack density, whereas rESWT was associated with increased microcrack length [30].

Several authors have described rESWT effects on skin surface temperature changes using thermography, which is a noninvasive imaging method that enables superficial heat emission to be detected, indicating surface temperatures [31]. Skin surface temperature evaluation provides valuable information for monitoring the physiological status of an animal, which is influenced by local metabolism, blood flow (vascular tone), perfusion in subcutaneous tissue, the metabolic activity of the muscles and coat thermal insulation properties [32,33,34]. Previous studies have highlighted the potential use of thermography for monitoring the rehabilitation effects of physical devices, such as laser and electromagnetic field therapy in healthy horses [35,36,37]. In a study presented by Lubkowska et al. [38] rESWT caused increased skin surface temperature in the infraspinatus muscle region immediately after treatment and 15 min after rESWT. However, 30 min after rESWT the skin surface temperature was similar to what it was before treatment. Another study presented by Verna et al. [39] evaluated skin surface temperature changes in the middle third of the dorsomedial aspect of the metacarpus and metatarsus bones in six untrained horses to determine the vascular effects of rESWT. However, the skin surface temperature of the treated limbs after rESWT did not differ significantly from the control limb. Other studies have assessed skin surface temperature changes after fESWT at the origin of the suspensory ligament and the fourth metatarsal bone in horses without lameness. The results showed no significant differences in mean surface temperature between the treatment and control limbs and no physiological effects after two fESWT treatments [40].

None of the previous studies mentioned above examined the effects of rESWT on clinically healthy muscle tissue, in terms of skin surface temperature or muscle tone changes. As such, the aim of the present study was to assess the influence of rESWT on skin surface temperature and longissimus dorsi muscle tone in clinically healthy racing thoroughbreds. The hypothesis of this study was that rESWT would lead to a significant increase in skin surface temperature. The longissimus dorsi area was chosen as an area of interest for the current study as young thoroughbreds undergoing race training have little subcutaneous fat tissue, which positively contributes to increasing the effectiveness of rESWT. According to a previous study, the subcutaneous fat depth for horses ranging between 1 and 5 years of age is 0.5 cm, while for thoroughbreds it is 0.6 cm. In addition, it has been confirmed that the longissimus dorsi muscle thickness in thoroughbreds is around 2.5 cm [41].

## 2. Materials and Methods

The Animal Welfare Advisory Team at Wrocław University of Environmental and Life Sciences approved the study design, in compliance with Polish and European Union legislation on animal experimentation (no 6/2023). The procedures used in this study were deemed not to cause pain, suffering, distress or lasting harm equivalent to, or greater than that caused by the introduction of a needle (article 1.5 f EU directive 2010/63/EU). Ethical approval was granted without a formal application. Written consent was obtained from Partynice Race Course in Wrocław regarding all the racehorses that participated in this study.

### 2.1. Animals and Data Collection

The research was performed on 24 2- to 3-year-old thoroughbreds. The sample size was based on a power calculation. All the horses had a similar level of fitness and were trained daily for flat racing at Partynice Race Course (Poland) during the 2022 season. They were trained by the same trainer and were housed in the same stable. All were healthy on clinical examination prior to the trial. The clinical history of each horse was obtained from case files to determine whether any of them had a history of back injuries treated by a veterinarian. Each horse was visually and manually assessed by an experienced equine clinician (P.Z.). In addition, a standard physical examination of the musculoskeletal system of each horse was performed to check for any clinical injuries. The examination included evaluated movement to identify the type and degree of lameness and to perform flexion tests [42]. The examination of the thoracolumbar region included palpation and mobility tests [43]. Included horses had longissimus dorsi muscle thickness measured ultrasonographically, the mean value of which was 2.43 cm (SD = 0.11 cm). The mean thickness of the subcutaneous fat was 0.5 cm.

The horses were divided into two groups: a study group of *n* = 12 (6 at an age of two years and 6 at an age of three years) and an rESWT-sham group of *n* = 12 (6 at an age of two years and 6 at an age of three years). Assignment of horses to the groups was performed randomly.

On the examination day, the study group underwent rESWT, while the rESWT-sham group was subjected to rESWT without activation of the probe. A thermographic examination, followed by palpation, was performed before treatment (BT) and just after treatment (JAT) in both groups to examine skin surface temperature changes overlying the longissimus dorsi muscle. The palpation examination was performed to assess the muscle tone of the treated area. Additionally, a thermographic examination was performed 10 min after treatment (10 AT).

### 2.2. Radial Extracorporeal Shock Wave Therapy

The IMPACTIS M rESWT apparatus (Astar, Bielsko-Biała, Poland) used on the treatment area had a compressor pressure of 1–5 bar. It generated 1–10,000 pulses between 1 and 25 Hz, generating 0.38 mJ/mm^2^ of energy on the surface of the transmitter. During treatment, each horse was held by a qualified person outside its box, in the stable corridor. The therapy was performed once unilaterally (left side) on the longissimus dorsi muscle, in the area between the fifteenth thoracic and the second lumbar vertebrae. The skin of the examination area was not clipped. The treatment area was 10 cm^2^ in size and was consistent for all the horses taking part in the experiment. Ten mL of ultrasound transmission gel was applied before rESWT, which was performed without sedation. The volume of the gel was measured using a 20 mL syringe. The parameters of the rESWT included 2000 shock waves with an energy flux density of 0.19 mJ/mm² and a frequency of 10 Hz. In each case, the probe was held in place with light pressure on the handpiece by the same equine clinician (K.Ś.). Pressure applied to the tissue being examined contributed to reducing the distance between the probe and the longissimus dorsi muscle, thus increasing the rESWT effect. Each treatment lasted for 3.33 min. The rESWT-sham group underwent the same procedure but with the device turned off.

### 2.3. Thermographic Examination

A calibrated VarioCam HR infrared camera (uncooled microbolometer focal plane array, Focal Plane Array sensor size of 640 × 480, spectral range 7.5–14 μm, noise equivalent temperature difference of <20 mK at 30 °C, using the normal lens with IFOV of 0.57 mrad, measurement uncertainty of ± 1% of the overall temperature range, InfraTec, Dresden, Germany) was used for thermographic examination. The standard thermographic examination protocol, as previously described by Soroko et al. [44], Soroko and Howell [45] and Zielińska et al. [46] was used. The horses were examined at rest, before their daily exercise. To minimize environmental influences such as air drafts and sunlight, the examination was conducted in an enclosed stable with closed windows. The ambient temperature in the stable was 17 ± 3 °C with a humidity of 50% (without major fluctuations), measured with a TES 1314 thermometer (TES, Taipei, Taiwan). Any dirt and mud present in the examination area was brushed away one hour before each examination. Prior to the examination, the horses underwent 20 min of acclimatization to the temperature in the stable. All the thermographic images of the back were taken from the dorsal aspect (Figure 1) and were performed by the same operator (M.S.-D.). The distance of the animal from the camera was set at 1.5 m for all images, with the emissivity set at 1 for all measurements [47]. The temperature was calculated manually using IRBIS 3 Professional software (InfraTec, Dresden, Germany), by one person (M.S.-D.), using the average temperature value in a rectangular area over the investigated areas [48].

### 2.4. Palpation Examination

The palpation examination was performed unilaterally on the longissimus dorsi muscle, in the area between the fifteenth thoracic and the second lumbar vertebrae on the left side of the back. The palpation scoring system for horse muscle tone was based on Varcoe-Cocks et al. [49]: (0) soft, with low muscle tone; (1) normal tone; (2) stiff but not painful; (3) stiff and/or painful with slight associated spasms, but without horse movement; (4) painful with associated spasms and local horse movement, i.e., pelvic tilt, extension response; (5) very painful with spasms and behavioral responses, i.e., ears flat back, kicking. The palpation scoring system was also used in our previous study [46] and an acceptable level of intra-observer repeatability for the grading scale was applied. The horses were examined and scored subjectively for longissimus dorsi muscle tone by a qualified equine physiotherapist (M.S.-D.) unaware of whether the horse in question had treatment.

### 2.5. Statistical Analysis

Statistical analysis was performed using STATISTICA v. 13.3 (TIBCO Software Inc. Palo Alto, CA, USA). The Wilcoxon pair test was used to determine whether there was a statistically significant difference between longissimus dorsi muscle tone before and after rESWT. The ANOVA test showed the mean skin surface temperatures BT, JAT and 10 AT in the test and rESWT-sham groups, as well as the results of significance tests and multiple comparisons (with Bonferroni correction). Spearman’s rank correlation coefficient and its 95% confidence interval were used to assess the strength and significance of the relationship between skin surface temperature and longissimus dorsi muscle tone.

## 3. Results

In both groups, there was an increase in skin surface temperature JAT and a decrease in temperature 10 AT below the initial value. In the study group, the skin surface temperature JAT was significantly higher than in the rESWT-sham group (*p* < 0.05). In the rESWT-sham group, the increase in skin surface temperature JAT was not significant (*p* > 0.05, Figure 2).

In the study group, a statistically significant reduction in longissimus dorsi muscle tone was observed JAT (*p* < 0.05). In the rESWT-sham group, the change in longissimus muscle tone was not significant (*p* > 0.05, Figure 3).

The correlation value between the skin surface temperature and the level of longissimus dorsi muscle tone for the study group was 0.350, while for the rESWT-sham group it was −0.102, statistically insignificant for both groups (*p* > 0.05, Table 1). In addition, the 95% confidence intervals for Spearman’s rho contain a value of zero, confirming the lack of correlation between the two variables.

## 4. Discussion

Our results showed a significant increase in skin surface temperature after rESWT in the study group, which confirmed our hypothesis. Evaluation of the skin surface temperature of the treated area indicated an increase in blood perfusion, associated with the increased metabolic activity present after rESWT. Similarly, a study based on rats showed increased muscular blood flow after application of ESWT [50]. However, Romeo et al. [51] have suggested that increased production of granulation tissue with leucocytes after ESWT is closely correlated with an increase in vascular density and local blood flow. The higher temperature thus represents increased blood flow in superficial veins [52]. Imboden et al. [53] have reported higher temperatures in the forelimb and hindlimb areas six hours after ESWT in horses with proximal palmar metacarpal and plantar metatarsal pain, but with the caveat that their results could have been affected by inflammation or a local anesthetic solution. Furthermore, other studies have reported that the average skin surface temperature of treated limbs after rESWT was not found to be significantly different when compared with the average temperature of control human limbs [38]. The horses that qualified for our study were clinically healthy and were not subjected to either sedation or local anesthesia. Factors which could have distorted our results, such as gel volume or the number of experimenters carrying out the procedure, were reduced to a minimum, which indicates that rESWT did in fact have a bearing on the increase in superficial skin temperature. Interestingly, horses without rESWT also showed an increase in body surface temperature just after rESWT. It is possible that the measured skin surface temperature may have been inaccurate, due to the camera’s non-perpendicular inclination to the surface of the horses’ backs. Furthermore, temperature increases in both groups may have been associated with the friction of the applicator upon the skin surface [38].

Additionally, a significant difference between longissimus dorsi muscle tone before and after rESWT was found in the study group. The method used in the present study clearly contributes to a significant reduction in muscle tone and pain sensation in the treated area. Studies on humans have shown ESWT pain-reducing effects in myofascial pain syndrome of the upper trapezius. Treatment significantly improves the verbal numeric pain scale, neck disability index and range of motion [54]. Moon et al. [55] conducted a study using fESWT in patients with subacute stroke, including ankle plantar flexor spasticity. Treatment was applied once a week for three weeks, at the musculotendinous junction of the medial and lateral gastrocnemius muscles. Clinical and biomechanical examinations were performed one week and four weeks after fESWT. Lower limb spasticity was found to have significantly improved immediately after fESWT and, additionally, was not significant at 1 week and 4 weeks after treatment. A similar study, to determine the influence of ESWT, was conducted on chronic stroke patients with extremity muscle tone [56]. Thirty patients were divided into ESWT and sham-ESWT groups. The results showed that muscle tone was higher in the sham-ESWT group than in the ESWT group, indicating that ESWT is effective in reduction of muscle tone.

Additionally, rESWT also contributes to improving the condition of treated tissue. Wang et al. [21] showed new capillary and muscularized vessels in subjects, which were present four and eight weeks after rESWT on the Achilles tendon–bone junction. However, the Santamato et al. [57] study on Achilles tendinopathy documented no neovascularization in 91.7% of patients after fESWT. Furthermore, they observed a reduction in blood vessels in some patients. Another study indicated a significant increase in blood pressure and partial oxygen pressure, as well as a decrease in carbon dioxide partial pressure, in patients with critical limb ischemia [58]. The obtained results contributed to an extension of walking distance without pain, with the effects remaining even after three months of treatment. Neovascularization may play a role in adequate blood supply, which can directly affect improvement in muscle tone.

In the present study, the correlation between temperature and muscle tone in the rESWT group was positive, but statistically insignificant (*p* > 0.05), perhaps as a result of the small sample size in the experimental design. There are presently no studies which can confirm our results regarding the correlation between skin surface temperature and muscle tone; thus, further research is necessary. In addition, palpation examination of muscle tone is limited by a large degree of subjectivity [49]; however, it is recommended by clinicians as it has an excellent interrater reliability among horses when using a categorical scoring system [49,59,60]. Ideally the subjective palpation examination should be replaced with pressure algometry as an objective method to determine pressure pain threshold by applying controlled pressure to the longissimus dorsi muscle. In addition, muscle palpation, which preceded the second thermographic examination, may have contributed to the increase in skin temperature observed. Therefore, further study should include control groups to determinate if palpation examination had an influence on the increase in skin surface temperature. Furthermore, this study included only 24 horses, which could have caused limitations in the statistical results.

Our results proved that rESWT has a thermal effect on healthy back tissue in horses. There are only a few other scientific studies confirming the effectiveness of rESWT. Our study has thus expanded knowledge in this area. Nonetheless, further work needs to be undertaken in order to determine the optimum number of treatment sessions and appropriate parameters to positively influence back disease in the long term.

## 5. Conclusions

This study confirmed the hypothesis that rESWT increases skin surface temperature overlying the longissimus dorsi muscle in clinically healthy racing thoroughbreds. Additionally, rESWT reduced the tone of the longissimus dorsi muscle. Further research is necessary in order to design shockwave treatment with appropriate parameters for effective and safe therapy. The results of our study serve as an impetus to expand future research in the field of equine rehabilitation.

## Figures and Tables

**Figure 1 animals-13-02028-f001:**
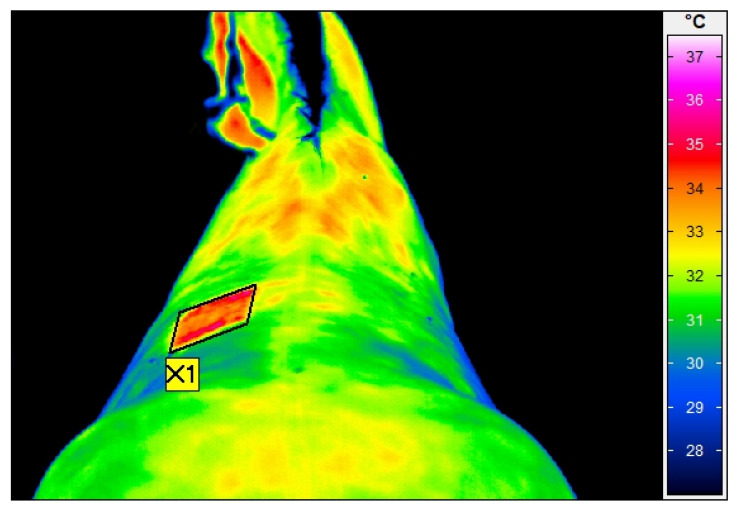
Thermographic image of the dorsal aspect of the back, taken just after radial extracorporeal shock wave therapy in the area of the longissimus dorsi muscle, between the fifteenth thoracic and the second lumbar vertebrae. The rectangular area (X1) shows the average skin surface temperature as 27.2 °C.

**Figure 2 animals-13-02028-f002:**
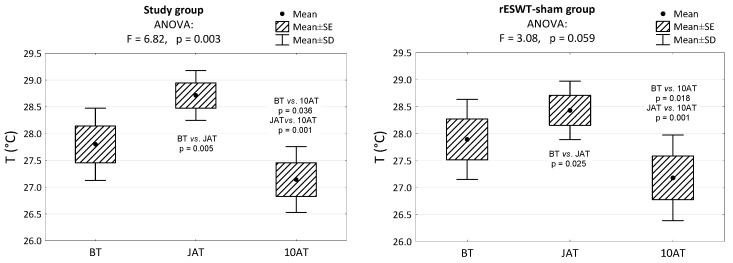
The average skin surface temperatures before treatment (BT), just after treatment (JAT) and 10 min after treatment (10 AT) in the study and rESWT-sham groups and the results of significance tests and multiple comparisons (with Bonferrini correction).

**Figure 3 animals-13-02028-f003:**
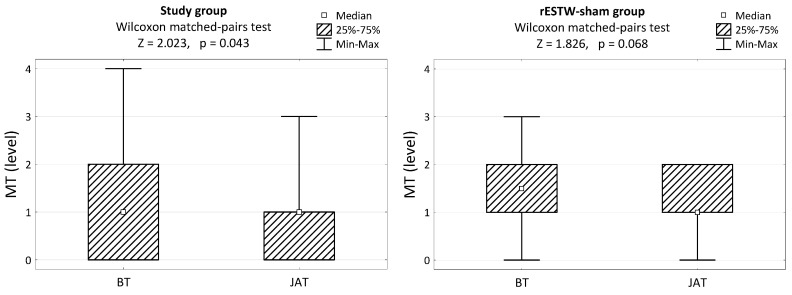
Longissimus dorsi muscle tone (MT) before treatment (BT) and after treatment (JAT) in the study and rESWT-sham groups and the results of significance tests.

**Table 1 animals-13-02028-t001:** Spearman’s rank correlation coefficient and its 95% confidence intervals, between the skin surface temperature and the level of longissimus muscle tone, in the study and rESWT-sham groups, before and after treatment.

Time	Group	
	Study	rESWT-Sham
Before rESWT, rho (95% CI)	0.427 (−0.194; 0.804)	−0.256 (−0.724; 0.373)
After rESWT, rho (95% CI)	0.467 (−0.145; 0.821)	0.258 (−0.371; 0.725)
Total, rho (95% CI)	0.350 (−0.062; 0.660)	−0.102 (−0.485; 0.314)

## Data Availability

The data presented in this study are available on request from the corresponding author. The data are not publicly available for privacy reasons.

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
