# Peer review of "The Effect of Radial Extracorporeal Shock Wave Therapy (rESWT) on the Skin Surface Temperature of the Longissimus Dorsi Muscle in Clinically Healthy Racing Thoroughbreds: A Preliminary Study"

_animals, 2023, doi:10.3390/ani13122028_

Round 1
Reviewer 1 Report
I regret to say that the article, although interesting, is not written in an accessible way. The introduction section gives the impression of being made up of sentences that lack readability. Although I am not qualified to assess the quality of English in this paper, I have a bad time reading the whole work linguistically. This problem can be easily solved and I am sure that the authors, in cooperation with a native speaker, will be able to cope with language fluency. However, in the presented work several major problems need to be addressed before further consideration of accepting the work for publication. Issues are listed in order of occurrence, not severity, but each of them is cause for at least major revision and even rejection and reconsideration after correction.
Firstly, the main question regarding the characteristics of rESWT is, what is the depth of penetration of the wave into the tissue? (add references) How deep was positioned the longissimus dorsi muscle under the skin? Please provide measurements for example ultrasound measurements of the subcutaneous fat plus skin thickness (SF-Skin) (https://doi.org/10.1186/s13028-016-0243-2) for each horse and the thickness of the longissimus dorsi muscle at the measurement area. Otherwise, it is difficult to prove that the wave in question reached the alleged area at all. It was initially mentioned in L 64-65 that "rESWT gives a more superficial effect, while fESWT, can penetrate to twice the distance of rESWT", however, the numerical data are required. Especially in the specific skin and fat tissues environment. The consequences of responding to this and the next remark should also be considered in the last paragraph of the introduction section (L 130-136).
Secondly, please see "palpation to assess longissimus dorsi muscle tone" L 17-19, L 32, L 167-169, L 208-218, and in other related places. The referred Varcoe‐Cocks et al. pressure algometry used to quantify muscle pain in racehorses was published in 2006. In the more recent works cited by you in the introduction, the muscle tone was measured using MAS (the Modified Ashworth Scale) (Gonkova et al., 2013; in spastic children, not horses; Marinelli et al., 2015; in humans, not horses). You also referred in the introduction section to the ESWT effect on the electrophysiologic properties of the gastrocnemius (Sohn et al., 2011), thus one may state that the measuring of the electrophysiologic properties of selected muscles is the reference method of the muscle tone measuring especially since sEMG has been introduced successfully to equine medicine. For this reason, I regret to say that the palpation-based muscle tone measurements you present can not be considered a reference and reflect normal muscle tone. Your research should be supplemented with electromyographic measurements in the vicinity of the examined area. Therefore, it is with great regret that I have to suggest the removal of the results related to the palpation measurement of muscle tension from the paper submitted for review. Moreover, detailed palpation of the imaging area 10 minutes before the thermographic examination may affect the results of the second thermographic imaging and measurement. However, I no longer want to undermine the credibility of the better part of your study. You should know it, especially since you referred to Soroko and Howell (2018), which highlight that "The minimum recommended acclimatization time for the horse before imaging is 20 minutes" and "The horse must have a clean, dry hair coat and skin and should be groomed at least 1 hour before the examination to eliminate artifacts arising from changes in surface emissivity", as the thermographic examination is very sensitive for an effect any external factors.
L 24, L 39, and L 314-316. I can not agree with the sentence "rESWT has a positive effect on the back muscles and potential to be used effectively in rehabilitation following back disease". Based on the presented preliminary research you can only conclude that after an rESWT session, the surface temperature of a small area of skin above a specific muscle (probably the musculus longissimus dorsi) increases, and the increase in surface temperature lasts for less than 10 minutes. You don't know if this effect is positive or negative for the (plural) back muscles. Based on the obtained results, you can not unequivocally state whether rESWT can be used in the rehabilitation of horses suffering from back diseases. You only examined healthy horses. Please, see that the effect described in L 19-20, L 33-34, and L 229-230 "In both groups, there was an increase in skin surface temperature just after rESWT and a decrease 10 minutes after rESWT" may be related to palpation, not rESWT. This limitation should be raised in the discussion section.
Please address the major and detailed comments and improve your work substantially, which I would love to see after resubmission.
In addition to the major comments, detailed remarks are provided below.
L 26 There is a lack of background sentences in the abstract section. Please, see that they are required.
L 30 "probe activation on the longissimus dorsi muscle" Please, correct this sentence. The probe does not touch the muscle.
L 31 "skin surface temperature changes" in the materials and methods section you don't know if you will get changes/differences. It's better to say compare... Please, rephrase.
L 37 Please, remove the p-value, as you did not support any other differences in the abstract section with the p-value. Please, also note that in the second part of this sentence "the rESWT-sham group was not significant", the subject is missing.
L 38 Is the sentence "There was no correlation between skin surface temperature and muscle tone" necessary in the abstract section?
L 47 Acoustic waves are a specific group of mechanical waves and are not identical. Please be precise.
L 51 Add a space mark between 5 and μs
L 53-54 Whether "These parameters depend on the level of energy, the pulse frequency, the type of tissue through which the acoustic wave propagates and the penetration depth" or "the level of energy, the pulse frequency, the type of tissue through which the acoustic wave propagates, and the penetration depth depend on wave characteristics/parameters"?
L 66 Please, detail the waves' effects on the target tissues. How do they depend on the properties of the tissue and do they vary by tissue type? skin, fat, connective tissue, muscles? The final biological effect, described in L 69-79 mainly concerns bones, bone marrow mesenchymal cells, and muscles, while you are mainly interested in skin, subcutaneous tissue, fat, fascia, and only ultimately muscle. Three of the four cited papers concern non-equine tissues, and the final thought, supported by the citation of Benson et al. (2007) research, indicates that rESWT does not affect the equine tissues studied. On the other hand, in the next paragraph the equine and human studies are mixed, so I can not follow which references refer to which species. This paragraph should be rewritten and combined with the following paragraph (L 80-95) into a shorter one to highlight the most important biological effects studied in horses for the target tissues of interest. The next paragraph, focused on neovascularization and vasodilatation, is better, however, still mixed. Moreover, you can not expect the neovascularization effect 10 minutes after a single application. Therefore, it would be worth focusing on vasodilatation. Also, note that the reference method for measuring vasodilatation and blood flow in horses is a Doppler ultrasound examination.
L 110-117 Do you need to enter half a paragraph about treating not what you are researching in horses? However, it is good that in horses, and not in mice or rats.
L 118-130 Here is what you need. In the introduction section, you should briefly describe what and how you studied, and in the discussion section, refer in detail to the numerical values and differences described by Da Costa et al. (2004), Lubkowska et al. (2018), Verna et al. (2005), and Ringer et al. (2005). But this is the right direction after all.
L 138-145 Approved or decided that the restrictions defined in the law shall not be applied? (following document no 6/2023 submitted as non-published materials. Are you sure this decision should be made by the Animal Welfare Advisory Team Supervisor and not the Chairperson of the Local Committee for Ethics in Animal Research? I believe this is a formal error and not intentional.
L 148 How the similarity of the fitness level was quantified and compared?
L 150 The interviews with the trainer (medical history?) can not be a reliable source of data for the diagnosis of any pathological disorders. Moreover, what do you mean by "any pathological disorders"? Please, rewrite to avoid confusion.
L 156-157 Were they divided randomly?
L 202 How the repeatability (between three measurements and between different horses) of rESWT and measurement of average temperature value in the rectangular area on horses back was ensured? Were anatomical reference points used? If so, which ones?
Reviewer 2 Report
Congratulations for your paper. The comments and suggestions for authors are attached.

Reviewer 3 Report
Manuscript: animals- 2342790
introduction
It would be useful to focus the introduction on the aim of the present study, and reformulate shock wave therapy description highlighting more recent advances and lack in current knowledge. Please rewrite and shorten the paragraph.
2.2. Radial Extracorporeal Shock Wave Therapy
were the horse manually restraint o put in a stock, during the therapy?
2.1. Animals and data collection
was the AAEP scale used for the orthopaedic examination of horses enrolled in the present study? please clarify
Table 1
it should be indicated in the table itself correlation coefficient and confidence intervals nomenclature, suggested under the time column.
Round 2
Reviewer 1 Report
I regret to say that the article, after resubmission, is still not written in an accessible way. In this revision, I will refer to my previous comments, indicating whether the raised problem has been resolved or not
1. Lack of readability. Recommended cooperation with a native speaker - not solved
2. Depth of penetration of the wave into the tissue - added
3. How deep was positioned the longissimus dorsi muscle under the skin? - partially. The authors, refer to Schauer 2015 (the citation format needs to be corrected in position 39 of the references), however, did not provide the measurement of subjected horses. Thus, it is still a speculation.
It is still difficult to prove that the wave in question reached the alleged area at all.
4. "rESWT gives a more superficial effect, while fESWT, can penetrate to twice the distance of rESWT" - the numerical data required - the numerical data were added, however, it is not clear if they refer to rESWT or fESWT.
5. The palpation-based muscle tone measurements can not be considered a reference - not solved - this is the major issue of the study which was completely ignored
6. Palpation of the imaging area before the thermographic examination may affect the results of the second thermographic imaging and measurement - partially - this sentence has been added, however, it does not solve the issue (suggests that your results are unreliable)
7. Replacing "rESWT has a positive effect on the back muscles and potential to be used effectively in rehabilitation following back disease" with "ESWT is effective in improving (reduction means improving?) muscle tone" still does not solve the issue of assessing muscle tone.
8. Based on the obtained results, you can not unequivocally state whether "rESWT can be used in the rehabilitation of horses suffering from back diseases" or that "increases skin surface temperature and reduction of the horse's response to back palpation play an important role in back muscle rehabilitation" - the problem with conclusions is still present
9. All limitations indicated in my first revision should be raised in the discussion section in a separate subsection. The authors should attempt structuring the discussion following a "common standard format" that usually consists of the following points:
a. One sentence summary that highlights the most relevant results.
b. A thorough discussion of each result obtained concerning the corresponding study objective: was the tested hypothesis confirmed or not? Why? What previous evidence supports the specific result or not? It is critical to compare/contrast the result obtained with previous literature in the equine species first, then in veterinary medicine, and finally in human medicine (if not enough data are available for comparison in veterinary medicine
c. Statement of study limitations
d. Future directions
Reviewer 3 Report
All comments have been aimed at and discussed.
Author Response
Thank you for your comment.